# Behavioural and Emotional Changes during COVID-19 Lockdown in an Italian Paediatric Population with Neurologic and Psychiatric Disorders

**DOI:** 10.3390/brainsci10120918

**Published:** 2020-11-27

**Authors:** Eugenia Conti, Giuseppina Sgandurra, Giacomo De Nicola, Tommaso Biagioni, Silvia Boldrini, Eleonora Bonaventura, Bianca Buchignani, Stefania Della Vecchia, Francesca Falcone, Caterina Fedi, Marisa Gazzillo, Gemma Marinella, Cristina Mazzullo, Jessica Micomonaco, Gloria Pantalone, Andrea Salvati, Gianluca Sesso, Valerio Simonelli, Greta Tolomei, Irene Troiano, Giovanni Cioni, Gabriele Masi, Filippo Muratori, Annarita Milone, Roberta Battini

**Affiliations:** 1IRCCS Fondazione Stella Maris, 56128 Pisa, Italy; eugenia.conti@fsm.unipi.it (E.C.); giuseppina.sgandurra@fsm.unipi.it (G.S.); giovanni.cioni@fsm.unipi.it (G.C.); gabriele.masi@fsm.unipi.it (G.M.); filippo.muratori@fsm.unipi.it (F.M.); 2Department of Clinical ad Experimental Medicine, University of Pisa, 56126 Pisa, Italy; tommaso.biagioni@fsm.unipi.it (T.B.); silvia.boldrini@fsm.unipi.it (S.B.); elelonora.bonaventura@fsm.unipi.it (E.B.); bianca.buchignani@fsm.unipi.it (B.B.); stefania.dellavecchia@fsm.unipi.it (S.D.V.); francesca.falcone@fsm.unipi.it (F.F.); caterina.fedi@fsm.unipi.it (C.F.); marisa.gazzillo@fsm.unipi.it (M.G.); gemma.marinella@fsm.unipi.it (G.M.); cristina.mazzullo@fsm.unipi.it (C.M.); jessica.micomonaco@fsm.unipi.it (J.M.); gloria.pantalone@fsm.unipi.it (G.P.); andrea.salvati@fsm.unipi.ir (A.S.); gianluca.sesso@fsm.unipi.it (G.S.); valerio.simonelli@fsm.unipi.it (V.S.); greta.tolomei@fsm.unipi.it (G.T.); irene.troiano@fsm.unipi.it (I.T.); 3Department of Statistics, LMU Munich, 80539 Münich, Germany; giacomo.denicola@stat.uni-muenchen.de

**Keywords:** neurologic disorders, neurodevelopmental disorders, psychiatric disorders, pediatric population, COVID 19, lockdown, mental health

## Abstract

On 11 March 2020, a national lockdown was imposed by the Italian government to contain the spread of COVID19 disease. This is an observational longitudinal study conducted at Fondazione Stella Maris (FSM), Italy to investigate lockdown-related emotional and behavioural changes in paediatric neuropsychiatric population. Families having children (1.5–18 years) with neuropsychiatric disorders referred to FSM have been contacted and proposed to fulfil two online questionnaires (General questionnaire and Child Behaviour Check List (CBCL)) to (i) compare (paired two-sample *t*-tests) the CBCL scores during lockdown with previous ones, and (ii) investigate the influence (multiple linear regression models) of variables such as age, diagnosis grouping (neurological, neurodevelopmental, emotional, and behavioural disorders) and financial hardship. One hundred and forty-one parents fulfilled the questionnaires. Anxiety and somatic problems increased in 1.5–5 years subpopulation, while obsessive-compulsive, post-traumatic and thought problems increased in 6–18 years subpopulation. In the regression models, younger age in the 1.5–5 years subpopulation resulted as “protective” while financial hardship experienced by families during lockdown was related to psychiatric symptoms increasing in the 6–18 years subpopulation. Some considerations, based on first clinical impressions, are provided in text together with comments in relation to previous and emerging literature on the topic.

## 1. Introduction

In order to contain the spread of SARS-CoV-2 disease, defined as a global pandemic by the World Health Organization (WHO), on 11 March 2020 a national quarantine was imposed by the Italian government until 3rd June. Recent research has shown that these lockdown measures, despite being mandatory to contain the spread of the infectious disease, have had a negative psychological impact on the general population and mental effects have been reported [1]. Previous studies reporting on the effects of quarantine in past epidemics and pandemics such as SARS, Ebola, the 2009 and 2010 H1N1 flu, Middle Eastern respiratory syndrome and equine influenza pandemics, were carried out in adult individuals [2,3,4,5,6,7,8,9,10,11], on healthcare workers [12,13,14,15], in school communities [16,17], on people who had been in contact with infected individuals [18,19], and on adult workers [20,21,22,23]. In these studies, mood deflection and an increase in depression and anxiety are the most frequently reported psychopathological features in the general adult population, while post-traumatic stress disorder (PTSD) was reported in health workers. Less is known about the effects of quarantine and isolation in the children and adolescents. Sprang and Silman 2013 [24] investigated the psychological effects after a pandemic disaster (H1N1, SARS) in children and their parents reporting that the disease containment measures can be harmful for a significant portion of children and their parents. Criteria for PTSD were found in 30% of isolated or quarantined children based on parental reports, and 25% of quarantined or isolated parents, based on self-report measures. Studies on the psychological and psychiatric effects of traumatic events on children and adolescents, such as earthquakes [25], tsunami [26], wars [27], and traumatic life events [28] have been carried out. The assessment measures were the Child Behaviour Checklist (CBCL) and the Youth Self Report (YSR), both worldwide [26,27,28] and in Italy, where these tools were used to evaluate the psychiatric symptoms in children exposed to the earthquake in L’Aquila [25].

Recent studies support the negative psychological impact of the COVID-19 epidemic quarantine on the adult population in China [29]. COVID-19 peritraumatic stress disorder was associated with gender, age, education, occupation and region of respondents. Indeed, females showed significantly higher psychological distress than males, and individuals between 18–30 years and over 60 years presented the highest levels of COVID-19 peritraumatic stress disorder. Moreover, those with a higher level of formal education were more distressed. However, there is little data available on the psychological consequences of lockdown in the paediatric population. In a recent article [30], the authors suggest the deleterious psychological impact of absence of school activity in healthy children and adolescents. Furthermore, the study interestingly showed an association between a greater expression of symptoms and a home delivery birth. Moreover, the eating dimension has been reported as altered in the young adult population of undergraduate students in a recent study [31]. Specifically, the authors report that the level of confinement-related stress increased the risk of problematic eating behaviours among students, especially in those presenting with eating-related concerns. To date, there are not available studies exploring the changes in the psychiatric symptoms in Italian children and adolescents referred for neurological and/or psychiatric disorders. The main aim of this study is to evaluate the possible changes in emotional and/or behavioural symptoms in patients referred to University Hospital IRCCS Fondazione Stella Maris (FSM) in relation to the COVID-19 lockdown (in terms of isolation, change of daily routine, interruption of school attendance and accessibility to health care). This evaluation was conducted by comparing the CBCL [32] assessment scores pre- and during lockdown in patients, ranging in age from 1.5–18 years old, who had presented with a complex neurological and psychiatric diagnosis, meaning with “complex” the high rate of comorbidities in the referred population. Second, the potential influence of variables such as age, diagnosis grouping, financial hardship have been investigated. These data were collected through a General Questionnaire developed in collaboration with the European Academy of Childhood Disability (EACD).

## 2. Materials and Methods

### 2.1. Study Design

This observational longitudinal study was conducted by a group of residents in the Department of Child Neurology and Psychiatry in the University of Pisa (Italy), settled in the Foundation Stella Maris (FSM), under the supervision of the Director of the School (RB). FSM is a Research and Care Institute admitting every year more than 3000 children and adolescents for diagnosis and treatment of neurological and psychiatric disorders. Moreover, it hosts the University residency program in Child Neurology and Psychiatry. The referring population is characterised by complex developmental problems, including psychiatric and neurological diseases with frequent comorbidities within and across the diagnostic categories.

In order to detect the potential emotional and behavioural changes in this population, the CBCL tool was uploaded on a devoted online platform (“REDCap”, Research Electronic Data Capture, Vanderbilt University, Nashville, TN, USA). The additional General Questionnaire set-up in collaboration with EACD was distributed to collect further environmental data.

The inclusion criteria for the study were as follows; (i) age under 18 years; (ii) presence of a neuropsychiatric disorder assessed in FSM by a multidisciplinary team (neurologist and psychiatrist, psychologist, speech therapist, educational therapist, and child therapist) and through the administration of standardised measures such as K-SADS-PL [33], ADOS-2 [34], ADI-R [35], CBCL [32] etc.; and (iii) availability of CBCL scores obtained before the lockdown between September 2019 and February 2020.

The families that matched the inclusion criteria were contacted by the residents by telephone and informed about the study. Those who were willing and keen to participate received instructions by e mail to complete the two questionnaires. Follow-up contacts with families in the enrolment-window were made and families who asked for help in the completion of the forms were assisted by phone. Recruitment lasted two weeks (from the 20 April to the 4 May), while lockdown was stopped by the Government on the 18 May). The study was approved by the Tuscany Paediatric Ethics Committee (N. 95/20- 20.04.2020).

### 2.2. Measures

#### 2.2.1. The General Questionnaire

The General Questionnaire was completed anonymously by the enrolled families. The same questionnaire was distributed in a further European study—“EACD COVID-19 Survey-Families”—which was conducted in parallel and promoted by the same association to evaluate the impact of the COVID-19 emergency on all European families with children or adolescents with neuropsychiatric disorders. Different aspects of the daily life of young patients and their families during the lockdown were investigated (EACD COVID-19 SURVEYS—INITIAL REPORT (2020) http://edu.eacd.org/eacd-covid-19-surveys-initial-report).

#### 2.2.2. Child Behaviour Check List (CBCL)

The Child Behaviour Check List 1.5–5 (CBCL 1.5–5) is a parent-report used to analyse behavioural symptoms in pre-schoolers. It presents 100 items describing the presence and the frequency of a specific behaviour through a three-point Likert scale (0, not true; 1, sometimes true; 2, very true). The CBCL 1.5–5 consists of 3 summary scales, 7 syndrome scales, 5 DSM-Oriented Scales, and 1 Stress Scale. The first are represented by Internalising, Externalising and Total Problems, the second by Emotionally Reactive, Anxious/Depressed, Somatic Complaints, Withdrawn, Sleep Problems, Attention Problems, and Aggressive Behaviour, the third by Affective Problems, Anxiety Problems, Pervasive Developmental Problems, Attention Deficit/Hyperactive Problems and Oppositional Defiant Problems.

The Child Behaviour Check List 6–18 is a 113-item parent report is used to assess child and adolescent psychopathology. Similarly, to the CBCL1.5–5, each item is rated on a three-point Likert scale. The items are categorised in 8 syndrome scales (anxious/depressed, withdrawn/depressed, somatic complaints, social problems, thought problems, attention problems, rule-breaking behaviour, and aggressive behaviour), 6 DSM-Oriented scales (Affective Problems, Anxiety Problems, Somatic Problems, Attention Deficit/Hyperactivity Problems, Oppositional Defiant Problems Conduct Problems), and 3 scales Scored Using T Scores for ASEBA Standard (Sluggish Cognitive Tempo, Obsessive-Compulsive Problems, Post-traumatic Stress Problems). The Syndrome Scale items are also classified in three summary scales: Internalising Problems, Externalising Problems and Total Problems.

In both CBCL (½–5 and 6–18) the score is considered clinically significant when the T-score is 63 or above for summary scales and 70 or above for syndrome, DSM-Oriented Scales and scales scored using T Scores for ASEBA Standard, while it is in the border clinical range when the value is between 60 and 63 for summary scales and between 65 and 70 for syndrome, DSM-Oriented Scales and scales Scored Using T Scores for ASEBA Standard. Values under 60 for the summary scales or under 65 for other scales are considered not clinically significant.

### 2.3. The Online Platform: REDCap

When the Italian Government recommended the public to isolate themselves at home and minimise face-to-face interactions, potential respondents were contacted by phone and invited to complete questionnaires in Italian. The survey was uploaded and shared on the REDCap online platform and the link to the electronic survey was distributed by means of invitations via e-mails. The survey included an introductory section describing the background and the aims of the study and information ethics for participants that could be downloaded at any time. All parents provided electronic informed consent. During the informed consent process, participants were informed that all data would be used for research purposes only. Respondents were also able to withdraw participation and abandon the survey at any stage before the submission process.

### 2.4. Statistical Methods

In order to extrapolate the emotional and behavioural changes, measures of T-Scores for the subscales included in the CBCL at two time-points (pre- and during lockdown) were obtained and compared through paired two-sample *t*-tests, whose use is justified through the central limit theorem.

To further investigate the differences between pre- and during lockdown T-Scores and break down the different sources of variability, we adopted a multiple linear regression model in which the difference in T-Scores (pre- and during) was used as response variable.

The covariates considered in the models are the weekly average hours of treatment pre-lockdown declared by the respondent, the age in years of the patient, the parents’ financial hardship during the lockdown period, and three not mutually exclusive diagnosis groupings: “Neurological Disorders”, “Emotional and Behavioural Disorders”, and “Neurodevelopmental Disorders”. In particular, in the present study the Neurological Disorder grouping includes patients presenting with cerebral palsy and epilepsy; The “Emotional and Behavioural Disorders” grouping includes patients presenting with generalised anxiety, mood disorders, obsessive compulsive and feeding disorders; finally, the Neurodevelopmental Disorders (NDD) grouping includes patients presenting with autism spectrum disorder, attention deficit and hyperactivity disorder, motor coordination disorder, intellectual disability, language/learning disorder. Due to the complex diagnosis of the patients enrolled in the study, a patient could simultaneously belong to one, two or three groupings. More specifically, the models were composed as follows,
μ_diff_ = β_0_ + β_1_ p_1_ + β_2_ p_2_ + β_3_ p_3_ + β_h_ hours + β_a_ age + β_f_ financial(1)

In this model, the response μdiff is the difference in a specific T-Score among the CBCL subscale scores before and after COVID-19 hit; p_1_, p_2_ and p_3_ are, respectively, “Neurological Disorders”, “Emotional and Behavioural Disorders”, and “Neurodevelopmental Disorders”; hours indicates the weekly average hours of treatment pre-lockdown declared by the respondent; age is the age (in years) of the patient; financial is a binary variable stating whether the family of the patient indicated that they experienced financial hardship during the lockdown period. Covariates p_1_, p_2_, p_3_, and age were obtained having studied the diagnoses in the IRCCS FSM clinical database. Covariates hours and financial were obtained from the EACD Questionnaire.

Sixteen differences in CBCL T-Scores were investigated as response variables for the 1.5–5 age group, while twenty differences were analysed for the 6–18 age group. Since the response is always calculated “pre- vs. during”, a negative difference indicates an increase in T-Score. Given this, covariates for which the effect has a negative sign are associated with an increase in T-Score and thus a worsening in the patient’s conditions, while the opposite is true for positive coefficients.

In the following section, coefficients and results with *p*-value lower than the conventional threshold for statistical significance of 0.05 will be underlined. Moreover, for an explorative purpose due to the size of the sample with regards to the magnitude of the measured effects, and due to the nature of the sample itself, results with *p* < 0.1 will also be highlighted.

## 3. Results

### 3.1. Enrollment and Population Description

Seven-hundred families were contacted by the residents of the School of Developmental Neuropsychiatry, University of Pisa. Six-hundred-and-twenty-one families filled out 75% of the EACD questionnaire, but only 494 completed the measures. Among these, 205 started the CBCL questionnaire according to the age of the patient and 201 completed it.

Only the fully compiled CBCL questionnaires were considered for the statistical comparison. Among the 201 CBCL questionnaires completed, 156 patients had had a previous CBCL assessment in our Institute in the previous six months, but 15 of these changed the CBCL questionnaire according to the age range. Consequently, the present paper includes the results of 141 subjects that filled in CBCL questionnaires useful for the comparison with those available in pre-lockdown. More specifically, 61 were completed by the parents of patients ranging in age from 1.5 to 5 years and 80 by parents of patients from 6 to 18 years old (Figure 1). As EACD questionnaire has been delivered also via Web, 47 families on our platform have completed EACD and CBLC questionnaires, but cannot be enrolled in this longitudinal study since they are not FSM patients and have not a previous CBCL assessment. These data will be used for other ongoing studies.

The sample 1.5–5 was composed of 8 females and 53 males. Forty-eight subjects presented overlaps in the three diagnosis groupings: 1 patient presented both Neurological and Emotional and Behavioural Disorders, 3 presented Neurological and Neurodevelopmental Disorders, and 30 presented both Neurodevelopmental and Emotional and Behavioural Disorders. Two patients presented Neurological Disorders and 25 presented with Neurodevelopmental Disorders. None of the children in this sample presented only Emotional and Behavioural Disorders (Figure 2a) or the co-occurrence of the all the three groups.

Sixteen females and 64 males constituted the 6–18 sample. Overlapping in diagnosis groupings emerged in 48 patients: 2 patients presented Neurological Disorders and Emotional and Behavioural Disorders, 2 patients presented Neurological and Neurodevelopmental Disorders, 41 presented Neurodevelopmental and Emotional and Behavioural Disorders. Three patients fit in all three groupings. Two patients presented Neurological Disorders, 20 presented Emotional and Behavioural Disorders and 10 patients presented Neurodevelopmental Disorders (Figure 2b).

### 3.2. Pre- vs. During Lockdown Comparisons

The means of the CBCL subscales T-Scores of all the patients were compared using *t*-tests. For the population aged 1.5–5, significant negative differences (that is a worsening in the clinical features) were observed in the Syndrome Scale Score in the Somatic Complaints (*p* < 0.1) and in the DSM-Oriented Anxiety Scale (*p* < 0.05). For the CBCL 6–18 questionnaire, a significant negative difference (that is a worsening) was observed in the Syndrome Scale Score Thought problems (*p* < 0.05) as well as in the Obsessive scale (*p* < 0.05) and the PTSD scale (*p* < 0.1). All the other CBCL domains in both forms did not show significant differences. A descriptive graphical representation of the significant T-Scores is given in Figure 3a,b.

The results obtained from the fitting of the multiple linear regression models, as in (1) in the statistical analysis section, are showed in Table 1 and Table 2, respectively, for CBCL 1.5–5 and CBCL 6–18. A detailed interpretation of the results can be found in the discussion section.

## 4. Discussion

To the best of our knowledge, this is the first study exploring emotional and behavioural changes during the COVID-19 lockdown in an Italian paediatric population with neurological and psychiatric disorders, using the CBCL as assessment measure. During the lockdown, the clinical activity of IRCCS Fondazione Stella Maris was strongly limited due to the emergency situation, but the residents immediately sought to guarantee via telephone contact whenever required. The quasi overnight change in the work routine created a unique opportunity to study the effects of this worldwide situation could have on children with neurological and/or psychiatric disorders.

It is known that lockdown is a potentially traumatic experience due to separation from relatives and friends, loss of freedom, uncertainty over disease status, and boredom. It can determine dramatic effects in the general population (as reviewed in [1]), while the consequences could even be worse in those families with severe concerns about the health of their children. As recently reviewed in [36], the COVID-19 outbreak was a particularly challenging time in terms of psychiatric symptoms occurrence, somatic symptoms, sleep disturbances for children and adolescents with special needs or disabilities, and low socioeconomic status [36].

Results from the current study shed light on difficulties experienced in our neuropsychiatric population, exploring the changes reported by their parents during this extreme social experience, by comparing pre-lockdown and during lockdown CBCL scores.

It is important to highlight that the population involved in the study is potentially more vulnerable than other neuropsychiatric patients as they present with complex comorbid diagnosis and are referred to the largest Developmental Neuropsychiatric Institute in Italy (FSM).

Despite these difficult conditions, the parents’ willingness to participate to the study and to explore behavioural and/or emotional changes during the lockdown was commendable and greatly appreciated, although some parents had difficulty completing the questionnaires as it was time-consuming. Moreover, the stressful situation experienced by families in terms of isolation, the closure of schools, worries over health and finances, loss of grandparents’ support, need to quickly adapt to “smart-working” at home, and ban on going outdoors to reach open spaces compounded the situation.

Furthermore, it is important to underline that the experience of specific worries about their child’s mental health increases the parents’ risk of depression, ruminative thought, anxiety, or other types of psychological distress, as reported in parents of children with autism spectrum disorders by [37]. Regarding families with children with autism spectrum disorders, stressful COVID-19-related consequences have also been recently reported by Colizzi and colleagues, using an online survey [38]. Despite the non-completion of the questionnaires by some parents, our research has managed to study a heterogeneous and complex population, ranging in age from 2 to 18 years, with neurological and psychiatric issues.

As far as the pre-school subgroup is concerned, significant worsening was found in the Syndrome Scale Score in the Somatic Complaints (*p* < 0.1) and in the DSM-Oriented Anxiety Scale (*p* < 0.05). This is not surprising, considering previous research that correlated anxiety disorders with somatisation in the pre-schooler population [39]. However, to the best of our knowledge, no data in this age population was analysed in COVID-19 studies, even if an increase in anxiety symptoms was reported in older children in several recent studies [40,41]. These results should be considered in a broader perspective, taking into account the relationship between the child and his/her caregiver and the child’s symptoms, as younger children are more vulnerable to maternal mental illness. As highlighted by the National Research Council and Institute of Medicine Committee (2009) [42], parental depression has been consistently found to be associated to early signs of “difficult” temperament in children, insecure attachment, emotional dysregulation, reduced cognitive and academic performances, cognitive vulnerability to depression, poor interpersonal functioning, and psychobiological dysfunction.

Besides, young children could be more sensitive to the spill over stress contagion as highlighted by [43] referring to COVID-19: the stress of increased work demands or financial burdens are likely to “spill over” into parents’ responsibilities as caregivers, and compromise their ability to provide sensitive and responsive care. Furthermore, the stress that children might experience as a result of changes in routine, such as being at home rather than at school, may “spill over” into how they interact with their siblings and parents. In a recent study carried out during the lockdown [44], both caregivers for children and adults with intellectual disabilities presented major levels of defeat/entrapment, anxiety, and depression which could affect the child–parent relationship.

As far as the schoolers subgroup is concerned (6–18 CBCL questionnaire), a significant negative difference was observed in the Syndrome Scale Score Thought problems, (*p* < 0.05), in the Obsessive scale (*p* < 0.05) and the PTSD scale (*p* < 0.1), thus meaning a clinical worsening during lockdown.

In a comprehensive review of studies conducted in children and adolescents during the COVID-19 lockdown [36], anxiety and depression were reported in most of the studies, while another study [45], exploring mental health in the community, reported a general intensification of the psychopathology, specifically in 2% of their population. Regarding the increase of Obsessive-Compulsive symptoms, this is in line with a previous study [46], which reported in children and adolescents during COVID-19 pandemic a worsening of obsessive compulsive disorder (OCD), namely in terms of contamination obsessions and cleaning/washing compulsions, using Children’s Yale-Brown Obsessive Compulsive (CY-BOCS [47]) and Clinical Global Impression-Severity Scale. It is of note that levels of Sensitivity and Specificity of Obsessive-Compulsive Scale from the CBCL were previously analysed and the results were found to be fairly high, demonstrating high diagnostic power according to favourable Positive-Predicted-value and Negative-Predictive-Value. A worsening in PTSD symptoms after COVID-19 hit has also been reported in a sample of young Chinese people with no neuropsychiatric issues [48]. The validity and factor structure of the CBCL-derived measures of trauma symptoms (PTSD) are supported in the literature, and the scale is useful to study trauma symptomatology in a sample of neglected children.

As far as the results of the regression models in this study, it has been investigated the effect on the changes between pre- and during lockdown scores in the CBCL scales of the variables financial hardship, age, weekly average hours of treatment pre-lockdown, and the previously defined diagnostic groupings. Considering the sub-population aged 1.5–5 years, even though some statistically significant effects show up in different regressions, no specific patterns seem to emerge in the comparison between the different diagnostic categories.

Almost all effects for the control variable age are negative, showing a tendency for the younger children in the 1.5–5 sample that could be interpreted as a better reaction to the lockdown. More sporadically within the Emotional and Behavioural grouping the dimension of “stress problems” and “pervasive developmental problems” appear significantly worsened (Table 1); not surprisingly, this finding could be related in this group to the coexistence of the Emotional and Behavioural disorder with Neurodevelopmental disorder (Figure 2a).

On the other hand, “anxiety problems” resulted increased in Neurological disorders grouping, possibly due to the familiar distress for the management of the organic issues (Table 1).

Regarding the sub-population aged 6–18 years, all estimated effects for “Emotional and Behavioural Disorders” have a positive sign, six of them (in the regressions for Internalising, Externalising, Total Problems, Affective, DSM ADHD and Attention difference in T-Scores) being statistically significant. This surprisingly suggests that there was tendency for a better reaction to the lockdown in these patients compared to those patients who do not present psychiatric problems (namely patients with neurological and/or neurodevelopmental disorders), controlling for all other factors. Note that, on average, patients with Emotional and Behavioural Disorders had higher starting T-Scores than the other patients in our sample, and thus a reduction in T-Scores for those patients could be interpreted as a regression of the scores of those patients towards the mean. Based on first clinical impressions, it can be speculated that this “protective effect” of having psychopathological diagnosis may be linked to the reduction of environmental requests during the lockdown situation. For example, the interruption of school and switch to online didactics could impact positively in the attention dimension [49]. Furthermore, due to the low exposure to competition and comparison with pairs, externalising and affective problems could have been reduced.

In addition, the results of the 6–18 CBCLs also show that financial hardship experienced by the family significantly worsened the following CBCL subscales; Total Problems, specifically Internalising Problems, the PTSD scale, the Obsessive-compulsive scale, and, in the Syndrome Scale Scores, the Withdrawn and Somatic problems subscales. This finding is in line with previous evidence [50], indicating that economic disadvantages were significantly associated with children’s internalising and externalising problems. Furthermore, a recent narrative review paper, focused on the COVID-19 pandemic burden on child and adolescent mental health [51], particularly underlined the string impact of financial losses. To our knowledge our results show scientific evidence regarding the crucial role of financial issues in managing a stressful family situation, especially in families with children and adolescents presenting with complex neuro-psychiatric problems, particularly regarding internalising problems.

Sporadically, it also appears that oppositional defiant dimension present high score during lockdown time when compared to Neurodevelopmental Disorders grouping, probably due to the loss of outdoor opportunities and the great amount of restrictions.

In the current study there are some limitations which are worthy of discussion, namely a possible selection bias. Indeed, only 201 out of the 700 families firstly contacted fully completed the two questionnaires. The limited number of participants that completed the CBCL could be due to the difficulties the families had understanding the online platform, where some instructions were written in English (an Italian translation was not possible for the software restrictions), and to the difficulty of completing a one hour questionnaire online. In addition, some families, who were contacted several times and were asked to complete the questionnaires, reported great difficulty in managing stressful home situation (e.g., taking care of children and adolescents on their own, working long hours at home). Moreover, the relatively low response rate could be a potential source of bias, since the parents that were able to fill out the whole questionnaire could be representative of a specific subpopulation with more facilities (for example open spaces, grandparents living with the families). Nonetheless, the results seem to be interesting for their internal validity. Gender bias represents another potential factor that has to be considered thus in literature the “sex biological variable” has been argued in neuropsychiatric field (e.g., [52]). In the current sample, the great majority of patients are male, and the small sample size does not allow to disentangle gender effect of pandemic-related symptoms changes.

In the missed 62 patients, due to the unavailability of pre-lockdown CBCLs or changes in the age group of patients over time, a longitudinal comparison was not possible. The sample size is thus certainly small compared to the size firstly estimated, and partially hindered the statistical significance of the results, albeit still sufficient for some preliminary analyses. A follow-up project to detect the medium-long term mental health consequences is ongoing. Furthermore, considering the importance of interaction between genetic and epigenetic factors in neuropsychiatric etiology, it would be interesting in future work to correlate available baseline genetic CNVs to pandemic-related symptoms augmentation, in order to investigate vulnerability and resilience.

## 5. Conclusions

This study represents a first step in the ongoing discussion regarding the emotional and behavioural consequences in a population with complex neurological and psychiatric disorders during such a uniquely stressful pandemic event. The present results are informative, both in preparation for a re-assessment of the clinical population after this period of isolation, as well as in the eventuality of a second wave of the pandemic, which may lead to a second lockdown.

## Figures and Tables

**Figure 1 brainsci-10-00918-f001:**
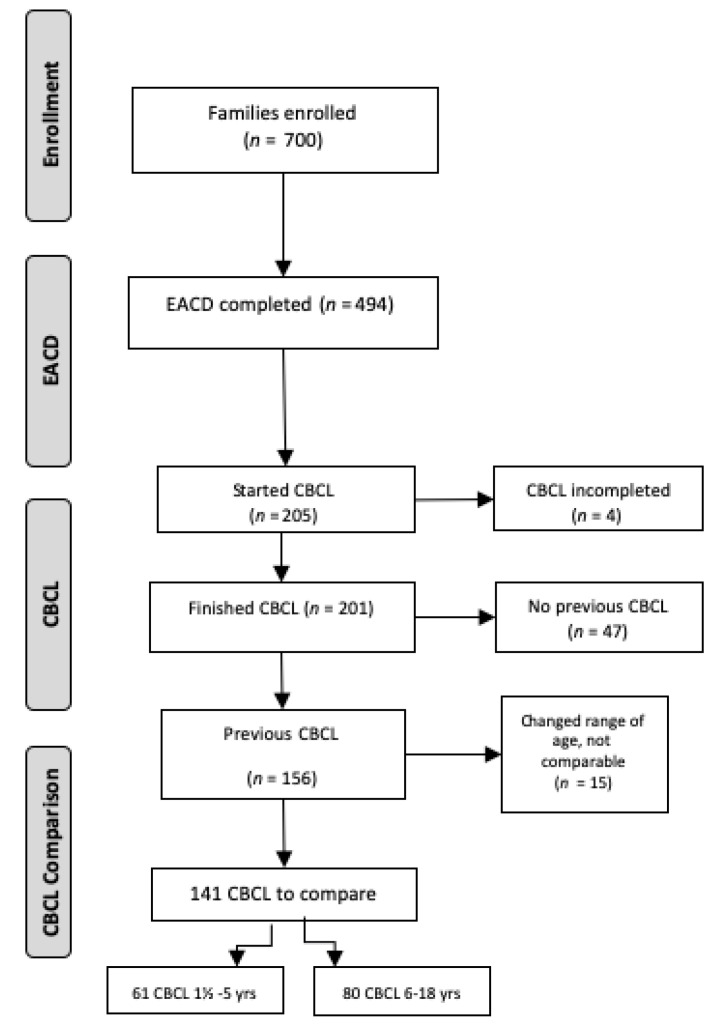
Patients Recruitment Flowchart. Abbreviations: EACD = European Academy of Childhood Disability; CBCL = Child Behaviour Checklist.

**Figure 2 brainsci-10-00918-f002:**
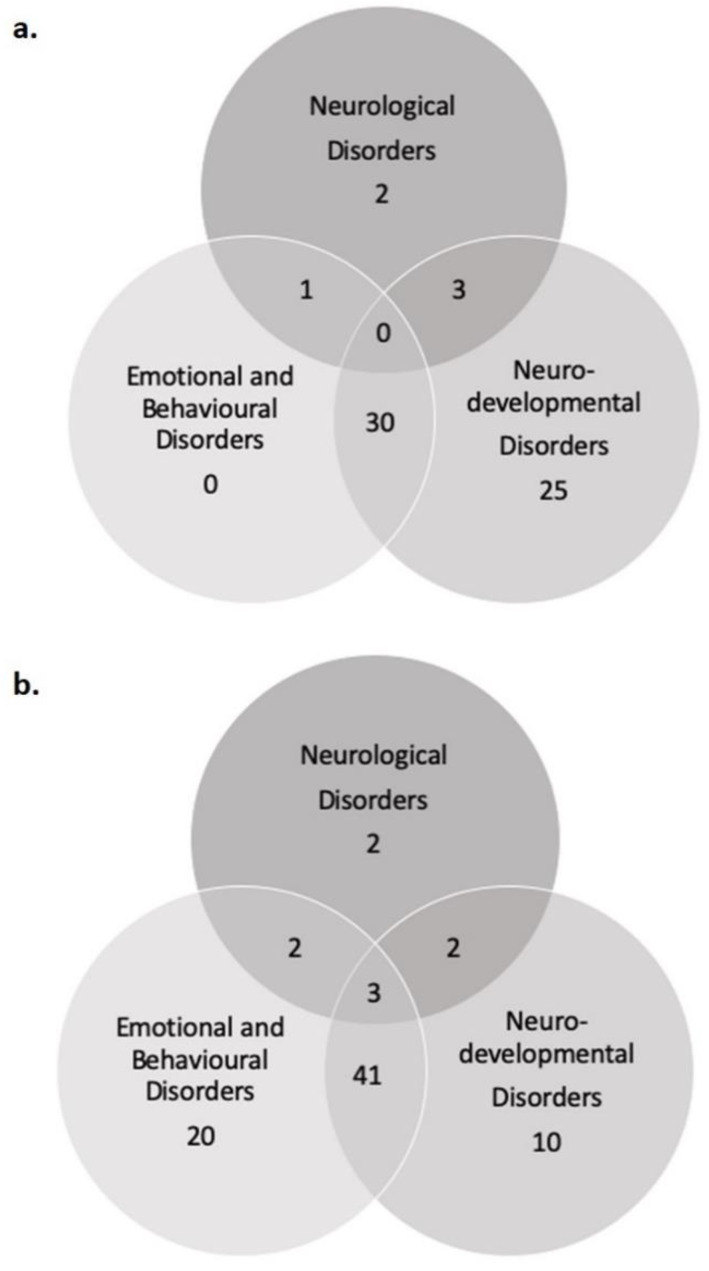
The distribution of patients is here shown across Diagnosis grouping for (**a**) CBCL—1.5/5 and (**b**) CBCL—6/18.

**Figure 3 brainsci-10-00918-f003:**
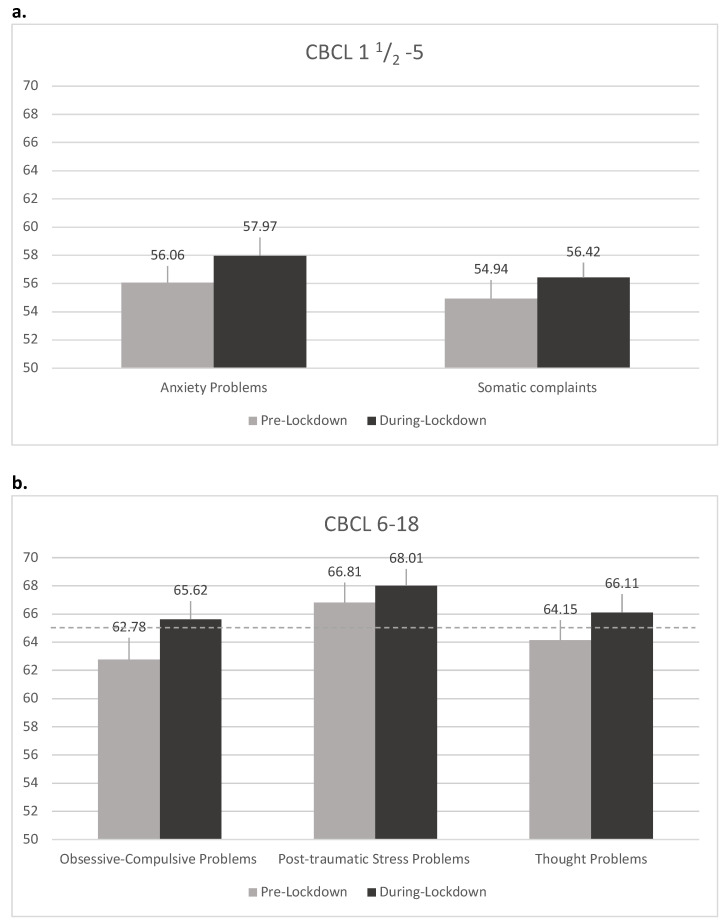
Student’s t–test results: (**a**) CBCL 1.5–5 years subpopulation. Graphical representation of the means in significantly different pre- and during lockdown T-Scores in the CBCL 1.5–5. The segment on top of each bar represents the upper half of the 90% confidence interval, while the dashed line indicates the clinical score for the subscales; (**b**) CBCL 6–18 years subpopulation. Graphical representation of the means in significantly different pre- and during lockdown T-Scores in the CBCL 6–18. The segment on top of each bar represents the upper half of the 90% confidence interval, while the dashed line indicates the clinical score for the subscales.

**Table 1 brainsci-10-00918-t001:** Regression models results (1.5–5 years).

	INT Problems	EXT Problems	Total Problems	Stress Problems	Affective Problems	Anxiety Problems	PDD	ADHP	ODP	Emotionally Reactive	Anxious/Depressed	Somatic Complaints	Withdrawn	Sleep Problems	Attention Problems	Aggressive Behaviour
**Intercept**	8.041	10.732	12.767	12.789	4.245	6.929	10.376	9.272	5.908	3.063	7.007	8.454	10.186	1.637	5.022	5.345
**Neurological Disorders**	−1.355	2.157	−0.996	−6.332	2.749	**−8.143 ****	2.009	−0.358	1.058	−1.502	−4.056	1.224	−3.306	3.130	2.673	2.579
**Emotional and Behavioural Disorders**	−3.353	−0.014	−1.655	**−4.599 ****	0.047	−2.462	**−5.19 ****	−0.875	1.153	−1.816	−1.017	0.363	−2.180	1.747	0.795	−0.037
**Neuro-Developmental Disorders**	−3.591	−7.699	−6.472	−7.410	1.368	−4.880	3.960	−4.781	−5.66	−1.073	−7.497	−5.359	−6.836	4.557	1.246	−5.045
**Treatment pre-lockdown (hrs/week)**	0.143	0.178	0.256	−0.257	0.346	−0.262	−0.162	0.529	−.0.21	0.440	−0.002	−0.245	0.217	0.187	0.066	−0.379
**Age**	−1.289	−1.245	**−1.976 ****	−0.903	−1.525	−0.566	−1.133	**−1.88 ****	−0.45	−1.046	−0.290	−1.264	−1.108	**−2.071 ***	**−2.079 ***	0.041
**Financial hardship during Covid**	1.884	0.857	1.329	0.561	−0.826	2.009	**3.607 ***	2.212	2.202	1.550	2.275	0.882	2.668	0.791	2.224	0.792

Schematic representation of the multiple linear regression models for CBLC 1.5–5. Each column represents a specific regression model, while rows contain the effects of covariates (Neurological Disorders; Emotional and Behavioural Disorders; Neurodevelopmental Disorders; hours; age; financial hardship during COVID-19). Statistically significant coefficients (* for *p* < 0.10, ** for *p* < 0.05) are highlighted in bold. Since the response is always calculated “pre- vs during”, a negative value indicates an increase in the CBCL Subscale T-Score and thus a clinical worsening.

**Table 2 brainsci-10-00918-t002:** Regression models results (6–18 years).

	INT P	EXT P	Total P	SCT	OCP	PTSP	Aff P	Anx P	Som P	ADHP	ODP	CP	AD	WD	SC	SP	TP	AT	RBB	AB
**Intercept**	−7.924	−1.836	−6.020	−0.650	−3.631	−4.046	−7.937	−5.329	−2.819	−6.012	2.747	−0.948	−6.748	−9.159	−1.503	−4.412	−3.357	−4.826	−3.017	−3.404
**Neuolog Disorders**	0.693	0.017	−0.435	−3.552	2.375	−0.299	−2.299	0.643	−1.555	1.593	−1.282	−0.856	0.452	0.187	−1.236	−1.097	−1.483	−1.276	1.278	−2.170
**Emotional BehaviourDisorders**	**4.091 ***	**4.020 ***	**4.658 ***	1.387	2.364	1.821	**3.843 ***	1.408	1.608	**3.784 ***	1.206	0.078	2.219	2.888	2.964	1.753	0.958	**4.756 ****	0.555	3.232
**Neuro-DevelopmDisorders**	0.725	−0.405	0.299	−1.982	−1.525	0.223	0.221	−0.376	2.182	2.651	**−3.203 ****	0.494	−0.262	2.956	0.135	0.511	1.110	−1.582	1.281	−1.168
**Treatment pre-LD (hrs/week)**	0.571	−0.040	0.298	0.008	1.125	0.481	0.322	**0.760 ***	0.290	0.049	0.396	0.054	0.681	0.319	0.219	0.470	−0.346	0.143	0.128	0.463
**Age**	**0.445 ***	−0.024	0.263	0.204	0.111	0.145	**0.539 ****	0.156	−0.015	0.040	−0.155	0.062	0.344	**0.619 ****	−0.058	0.184	0.249	0.193	0.103	0.066
**Financial Hardship**	**−3.808 ****	−1.203	**−2.959 ****	−1.186	**−3.248 ***	**−2.540 ***	−1.288	0.409	−1.615	−1.497	−0.743	−0.239	−1.706	**−4.523 ****	**−3.340 ****	1.149	−2.644	−1.866	−0.038	0.527

Schematic representation of the multiple linear regression models for CBCL 6–18 results. Each column represents a specific regression model, while rows contain the effects of covariates (Neurological Disorders; Emotional and Behavioural Disorders; Neurodevelopmental Disorders; hours; age; financial hardship during COVID-19). Statistically significant coefficients (* for *p* < 0.10, ** for *p* < 0.05) are highlighted in bold. Since the response is always calculated “pre- vs during”, a negative value indicates an increase in the CBCL Subscale T-Score and thus a clinical worsening.

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
