# Peer review of "Behavioural and Emotional Changes during COVID-19 Lockdown in an Italian Paediatric Population with Neurologic and Psychiatric Disorders"

_brainsci, 2020, doi:10.3390/brainsci10120918_

Round 1

Reviewer 1 Report

In this manuscript, Conti et al., investigated the COVID-19 imposed lockdown-related emotional and behavioral changes in the pediatric Italian population (1.5-18 yrs) with the existing neuropsychiatric population. The study is very relevant in the current state of the world and we need more studies like this. The impact of the lack of social interaction and isolation in children is unchartered territory and one that needs more attention.

The manuscript is well written with a detailed introduction, and the methods are clearly explained and the results are discussed in a clear and concise manner.

A few comments:

  • Although the study started with a promising 700 families, it narrowed down to 201 subjects, out of which only 141 patients had information from pre-lockdown. So, the final count is quite less and would need more subjects to validate these results. The authors mention that 494 families completed the questionnaire but 289 did not fill in the age. Can they go back to the families and confirm the ages? Or can they use this data with a larger age range? This would significantly increase the validity of the study.
  • The number of female subjects (8 females in 1.5-5 yrs age range, 16 females in 6-18yrs) is very less compared to the males (53 males in 1.5-5 yrs age range and 64 males in 6-18yrs). While I understand the limitations of the study, the authors should discuss this in the discussion with relevant reference to literature about the sex bias in neuropsychiatric diseases and how this may potentially impact the effect of the lockdown. Future studies should be conducted with equal representation of both sexes.
  • Since many of the neuropsychiatric disorders are genetic, I wonder if the authors have similar information about these children’s parents and siblings (if any). Even if it is for a few families, this information would be quite significant. Or at least they can discuss this in the discussion and follow it up in a future study.

Author Response

In this manuscript, Conti et al., investigated the COVID-19 imposed lockdown-related emotional and behavioral changes in the pediatric Italian population (1.5-18 yrs) with the existing neuropsychiatric population. The study is very relevant in the current state of the world and we need more studies like this. The impact of the lack of social interaction and isolation in children is unchartered territory and one that needs more attention.

The manuscript is well written with a detailed introduction, and the methods are clearly explained and the results are discussed in a clear and concise manner.

We thank the reviewer for these positive comments on the work.

A few comments:

  • Although the study started with a promising 700 families, it narrowed down to 201 subjects, out of which only 141 patients had information from pre-lockdown. So, the final count is quite less and would need more subjects to validate these results. The authors mention that 494 families completed the questionnaire but 289 did not fill in the age. Can they go back to the families and confirm the ages? Or can they use this data with a larger age range? This would significantly increase the validity of the study.

Certainly, the expectation of response rate was higher at the beginning of the study. However, we wanted to report this data faithfully, in order to have the possibility to report in the discussion section the difficulties that many parents have explained to residents during the phone-contacts, thus beeing very busy in their complex children daily care. Two hundred and eighty-nine families have not been counted in the current study since they fulfilled only the first questionnaire (EACD questionnaire) but are involved in the EACD European Survey Study (In preparation). Fifteen children were not enrolled in the longitudinal analysis because they changed CBCL form due to age (CBCL ½-5 (2019-2020) vs CBCL 6-18 (lockdown period).

  • The number of female subjects (8 females in 1.5-5 yrs age range, 16 females in 6-18yrs) is very less compared to the males (53 males in 1.5-5 yrs age range and 64 males in 6-18yrs). While I understand the limitations of the study, the authors should discuss this in the discussion with relevant reference to literature about the sex bias in neuropsychiatric diseases and how this may potentially impact the effect of the lockdown. Future studies should be conducted with equal representation of both sexes.

We agree with the reviewer. A comment on this has been added to the final part of the discussion, within the limitation considerations. A recent reference on the topic has been added, too. (Lines 435-439)

  • Since many of the neuropsychiatric disorders are genetic, I wonder if the authors have similar information about these children’s parents and siblings (if any). Even if it is for a few families, this information would be quite significant. Or at least they can discuss this in the discussion and follow it up in a future study.

Certainly, “genetic x environmental factors” are crucial in neuropsychiatric etiopathogenesis. The majority of patients followed at IRCCS FSM undergo genetic screening. We agree with the reviewer that this must kept into consideration in the following studies.

We have added a comment in the discussion on this. (lines 445-447)

Reviewer 2 Report

This article reports a study comparing the emotional and behavioral reactions of Italian children suffering from psychiatric and neurological pathologies during the confinement of spring 2020. The reported study is observational, with a first measurement carried out before the confinement and a second one during, through self-reported measurements and carried out by families and parents. A set of covariates were measured that relate to demographic variables. In itself, these data are interesting. The article is well-written, although the discussion is too lengthy when compared to the data and study design. The statistical analyses are conducted properly, but the choice of keeping the results at p<.10 is in my opinion questionable, especially given the relatively small sample size (understandable in view of the population studied).

In my opinion, the article needs a few modifications. You will find more detailed remarks below:

  • In the introduction, the authors can also cite Flaudias et al (2020) on the effects of confinement on stress and eating disorders in a student population in France.
  • Can the authors define what they mean by "complex neurological and psychiatric diagnosis"? Since the effects of confinement are supposed to manifest themselves on relatively precise dimensions (i.e., depression and anxiety), it seems to me that this population is likely to mix individuals with different baseline levels on these two dimensions.
  • I understand the model well and the statistical approach is quite justified in my opinion, but would it be possible for the authors to adopt a more readable writing of regression models? Maybe the problem comes from me, but the symbol "*" is usually used to denote an interaction between two variables, not the relative slope has a covariate.
  • With regard to the choice of thresholds, the opposite reasoning should be applied. As n increases, the standard error decreases and the precision of the effect size estimate increases. The risk is therefore that extremely small differences are "artificially" revealed when the sample is large, but this does not mean that lower alphas should be taken when the sample is small, since the standard error, and hence the precision of the estimate, is low.
  • The final sample size should be clearly announced somewhere in the results section.

Author Response

This article reports a study comparing the emotional and behavioral reactions of Italian children suffering from psychiatric and neurological pathologies during the confinement of spring 2020. The reported study is observational, with a first measurement carried out before the confinement and a second one during, through self-reported measurements and carried out by families and parents. A set of covariates were measured that relate to demographic variables. In itself, these data are interesting. The article is well-written, although the discussion is too lengthy when compared to the data and study design. The statistical analyses are conducted properly, but the choice of keeping the results at p<.10 is in my opinion questionable, especially given the relatively small sample size (understandable in view of the population studied).

We agree with the Reviewer that the choice of including the results between .05 and .10 could be questionable and uncommon. For these reasons, in the MS we have reported and considered separately the results significant at <.05 and those between .05 and .10. We think that for and explorative purpose, the results with p between 0.05 and 0.10 could be justified by the fact that, as stated in the paper, the results are mainly interesting due to their internal validity, and coefficients estimates are to be interpreted as general within-sample tendencies. Given this, and since the magnitude of the effects is rather small relatively to the sample size, we think that it is useful to consider coefficients with p between .05 and .1, while at the same time keeping them distinguished from those with p<.05, allowing the reader to have a more complete picture of the tendencies showed.

In my opinion, the article needs a few modifications. You will find more detailed remarks below:

  • In the introduction, the authors can also cite Flaudias et al (2020) on the effects of confinement on stress and eating disorders in a student population in France.

Thanks for the suggestion. Now, this article is cited in the introduction although the suggested article regards young adult population.

  • Can the authors define what they mean by "complex neurological and psychiatric diagnosis"? Since the effects of confinement are supposed to manifest themselves on relatively precise dimensions (i.e., depression and anxiety), it seems to me that this population is likely to mix individuals with different baseline levels on these two dimensions.

We decided to use the label “complex neurological and psychiatric” in order to explain the complexity of patients that usually refer to the University and Research Institute as IRCCS Stella Maris Foundation. A specification has been added in lines 86-87

  • I understand the model well and the statistical approach is quite justified in my opinion, but would it be possible for the authors to adopt a more readable writing of regression models? Maybe the problem comes from me, but the symbol "*" is usually used to denote an interaction between two variables, not the relative slope has a covariate.

It is right. The reviewer highlighted the presence of * that is a typo.

  • With regard to the choice of thresholds, the opposite reasoning should be applied. As n increases, the standard error decreases and the precision of the effect size estimate increases. The risk is therefore that extremely small differences are "artificially" revealed when the sample is large, but this does not mean that lower alphas should be taken when the sample is small, since the standard error, and hence the precision of the estimate, is low.

We agree with the reviewer; our data need to be interpreted with more carefulness.

As stated above, our choice of having two separate thresholds is related to the explorative aim of the results interpretation. 

  • The final sample size should be clearly announced somewhere in the results section.

We completed the information requested. See lines 213-214

Reviewer 3 Report

Although former studies have been conducted on this topic, this article contributes to bring important informations on the consequences of lockdown in pediatric population followed up for neurological or psychiatric disorders .

The article is clear. The conclusions are in line with the objectives.

As stated in the discussion a limitation of this study is a potential selection bias. It is said that FSM admits every year more than 3000 children and adolescents. It is not clear which criteria have been chosed to select the 700 contacted families. Are they only inclusions criteria ?

Another limitation is the date on which  families have been contacted after the lock down. Is there  a difference in the date of interview by age or disorders ? The duration of lock down could have an  influence on the answers.  Moreover, the duration between the two questionnaires could also have a test retest influence.

Author Response

Although former studies have been conducted on this topic, this article contributes to bring important information on the consequences of lockdown in pediatric population followed up for neurological or psychiatric disorders.

The article is clear. The conclusions are in line with the objectives.

We thank the reviewer for the positive comment on the importance of the topic faced.

  • As stated in the discussion a limitation of this study is a potential selection bias. It is said that FSM admits every year more than 3000 children and adolescents. It is not clear which criteria have been chosen to select the 700 contacted families. Are they only inclusions criteria?

We do apologize if we did not well explain this point. The 700 hundred families have been selected to be    contacted by phone, having access to our CBLC data base (see Inclusion criteria). We choosed a period from September 2019 to February 2020 in order to reduce the potential bias due to “time” to the second CBCL Assessment.

It is true that in the flow chart is reported that some patients have been ruled out of the current analysis since they had not a previous CBCL assessment. This is due to the fact that EACD survey (the first questionnaire) has been delivered through the web and some families (N  = 47), out of our phone contacts, have completed also CBCL (second questionnaire). This data will be used for future studies.

A clarification has been added in the text. (Lines 216-219)

  • Another limitation is the date on which families have been contacted after the lock down. Is there a difference in the date of interview by age or disorders ? The duration of lock down could have an influence on the answers. Moreover, the duration between the two questionnaires could also have a test retest influence.

At this point there is a misunderstanding. We do apologize. We did not extend recruitment out of lockdown. We went through the text and specified range date of recruitment. (Line 119-120)